# Ultrawideband Precision RCS Regulation for Trihedral Corner Reflectors by Loading Resistive Film Absorbers

Tianqi Sun [1], Fangfang Yin [1], Chengxiang Xu [1], Shan Zhao [2,*], Hua Yan [3] and Hongcheng Yi [3]

1 State Key Laboratory of Media Convergence and Communication, School of Information and Communication Engineering, Communication University of China, Beijing 100024, China
2 School of Information Engineering, Beijing Institute of Graphic Communication, Beijing 102600, China
3 Science and Technology on Electromagnetic Scattering Laboratory, Beijing 100854, China
* Correspondence: shanzhao_wdbk@163.com

**Abstract:** In this paper, an absorber with a resistive film is proposed, which holds stable polarization absorption characteristics at large angles by exploring the principle of absorption structure. The absorber contains three layers: the top layer is a metal patch with a resistive film, and the middle layer is covered with a metal patch. The thickness of the absorber structure is only 0.5 mm, and the absorber can be perfectly attached to the trihedral corner reflector (TCR) without affecting the characteristics of the reflector. By laying the absorber on the TCR, the TCR is stabilized at 8–40 GHz in the range of the incident angle ±20°, and the radar cross-section (RCS) reduction value of the TCR fluctuates around 1 dBsm. Thus, the RCS values of the three frequency points of 10 GHz, 15 GHz, and 35 GHz are consistent to achieve the accurate regulation of RCS. The stability of the absorber at different frequencies can be achieved, which is essential for the precise RCS regulation of complex scatterer structures. The effectiveness of the method was verified in experiments using a TCR metamaterial absorbing unit with resistive film.

**Keywords:** trihedral corner reflector; resistive film; RCS regulation; angle insensitive; absorber; metamaterial

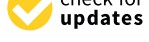



## 1. Introduction

In contemporary detection technology, the detection band has expanded from visible light to infrared and radar bands, which poses a challenge to camouflage technology [1]. In addition, the multi-band detection trend that has emerged in recent years requires compatible camouflage technology. TCR is a trihedral structure composed of three metal plates, often used to camouflage aircraft, missiles, and ships. Trihedral corner reflectors have large backscattering RCS over a considerable angular range. They do not require any power supply, are mechanically easy to construct, and operate under challenging conditions [2,3]. Additionally, TCR is deployable [4,5]. It is precisely because of the backscattering characteristics that it is difficult to camouflage under an extensive angle range and multiple frequency points.

In modern high-tech warfare, with the rapid development and broad application of reconnaissance surveillance and precision guidance technology, camouflage protection has become a vital prerequisite for coping with reconnaissance threats and improving survivability. Disguise is a variety of covert measures taken to deceive or confuse the other party to conceal the truth and reveal the falsehood; it is an integral part of the military's combat guarantee. Camouflage technology is the most critical technology for developing military power under the conditions of information warfare [6–9]. In passive radar interference, it is essential to realize the flexible regulation of electromagnetic scattering. The research on traditional corner reflectors is for fixed shapes; RCS varies with frequency, which limits its use, and the types of camouflage deception are limited, and camouflage a single type of

target. Therefore, the researchers achieved electromagnetic camouflage by combining the corner reflector with the metasurface to enhance and weaken the RCS of the corner reflector.

In passive radar interference, electromagnetic metasurface technology has essential application prospects for realizing the flexible control of electromagnetic scattering. Absorbers can be used to control electromagnetic waves; absorbers have been developed to reduce the size and improve the absorption performance [10,11], but they usually have bandwidth limitation problems. Several absorber designs have also been proposed to increase the bandwidth [12–15], but the variation in the incident angle within this resonant frequency range has not been considered. Arranging the metamaterial absorber in a dendritic pattern [16] can cleverly combine dielectric and ohmic losses. Angular stability can be achieved under the measurement of incident angles from 0° to 60°, but its wider bandwidth still needs further study. Another approach to broadband absorbers is to use resistive films printed on dielectric substrates [17–20]. The resistive film structure has a relatively stable equivalent impedance, and its replacement of the metal structure is expected to broaden the absorbing frequency band. At the same time, the resistive film structure has substantial polarization insensitivity and overall incident angle stability; therefore, using a resistive mode structure is a good choice.

The organization of this article is as follows. Section 2 describes the proposed theoretical formula for regulating the backward RCS of the triangular TCR and concludes that it is necessary to load an absorbing element on the TCR to realize the precise regulation of its backscattering RCS. In Section 3.1, the design of absorbers based on resistive film technology is proposed, and the precise and effective regulation of the electromagnetic resonance characteristics of the absorber can be achieved by rationally designing new structural units. Section 3.2 presents the experimental results and compares them with the simulation results. Finally, the conclusions are discussed in Section 4.

## 2. Operating Principle

To realize the regulation of TCR backward RCS, it is necessary to deduce the theoretical formula of backward RCS reduction. The GO (geometrical optics) method is a simulation method for the high-frequency limit case with zero wavelength. The scattering phenomenon at this time can be treated as a classical ray tracing, which follows Snell's reflection law. Radar scattering cross-section can be defined as:

$$RCS = \lim_{R \to \infty} 4\pi R^2 \frac{S_r}{S_i} = \lim_{R \to \infty} 4\pi R^2 \left| \frac{\vec{E}_r}{\vec{E}_i} \right|^2 = \lim_{R \to \infty} 4\pi R^2 \left| \frac{\vec{H}_r}{\vec{H}_i} \right|^2 \tag{1}$$

where $S_r$, $E_r$, and $H_r$ represent the scattered wave energy flow density, scattered electric field, and scattered magnetic field at the radar receiver, respectively; $S_i$, $E_i$, and $H_i$, are represented as the energy flow density, incident electric field, and incident magnetic field of the incident wave at the target, respectively. Compared with pure metal surfaces, the RCS reduction value of an equally sizeable metasurface can be expressed as:

$$\sigma_R = 10 \log \left( \frac{\lim\limits_{R \to \infty} 4\pi R^2 \left| \frac{\vec{E}_r}{\vec{E}_i} \right|^2}{\lim\limits_{R \to \infty} 4\pi R^2 (1)^2} \right) = 10 \log \left| \frac{\vec{E}_r}{\vec{E}_i} \right|^2 = 10 \log |S_{11}|^2$$

$$= 20 \log |S_{11}| = |\Gamma| \tag{2}$$

where $|\Gamma|$ is the $|S_{11}|$ of the reflection coefficient of logarithmization. The above formula shows that compared with the same large metal surface, the amount of RCS reduction in the metasurface is independent of its size, only related to the reflection coefficient, and the RCS reduction is equal to the magnitude of the reflection coefficient in terms of value.



Converting the above formula into the RCS reduction value corresponding to different frequency points of logarithmic operation can be expressed as:

$$\Delta\sigma_R = 20\lg\frac{E_r}{E_i}(dB) \tag{3}$$

Equation (3) shows that the RCS change is independent of its size and effective reflection area and is only related to the incident electric field and scattered electric field, i.e., to the reflection coefficient on each surface of the TCR. For the TCR composed of a metal plate with an edge length of a = 200 mm, the backward RCS is also different at different incident angles. The metal TCR was simulated and calculated using CST simulation software. The excitation type we set was the plane wave simulated in the time domain solver, using the RCS value of the probe to detect. The simulation results of the backward RCS detected by the probe at different incident angles can be obtained, as shown in Figure 1.

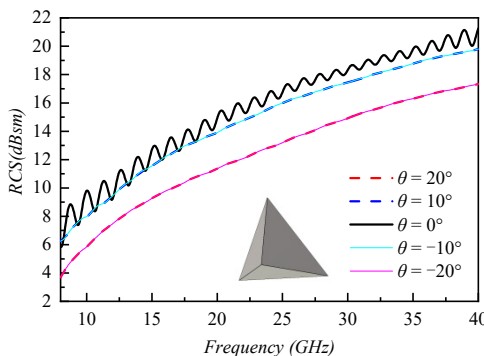

**Figure 1.** Variation in backward RCS with a 200 mm edge length of TCR θ offset by ±20°.

It can be seen from Figure 1 that the net reductions in TCR backward RCS at 15 GHz and 35 GHz (compared with the backward RCS value at 10 GHz, keeping the RCS value of 10 GHz as unchanged as possible) are 2.93 dB and 9.99 dB, respectively, regardless of the incident wave angle. In order to achieve the same backward RCS at 10 GHz, 15 GHz, and 35 GHz, it is necessary to obtain the reflectivity required by the unit at multiple frequency points according to the reduction value. In summary, to complete the precise control of the RCS of the backscattering of the broadband multi-band TCR, the absorbing unit loaded on the angular inversion needs to have a stable absorption rate in the wide-angle incident range.

## 3. Structure Design and Simulation Results

### 3.1. Design of Metamaterial Surface Based on Resistive Film Technology

According to the symmetry of the TCR model, if the absorbing unit does not have the ability for polarization conversion, the ratio of the incident wave's horizontal and vertical polarized waves on each surface should be 1:1. The surface element of the TCR has a reflection coefficient of ($\Gamma = \Gamma_{TE} + \Gamma_{TM})/2$ per face. If the three reflection coefficients are all equal, the total reflection coefficient is three times the reflection coefficient. If the RCS of different frequency points of each surface is reduced to about 1 dB and 3 dB, the absorption rate is too low and accurate for an absorber using metal resonance. The commonly used absorbing structures cannot stably complete the absorbing tasks of specific proportions under the premise of significant angle changes.

The absorber cell structure based on resistive film technology is shown in Figure 2. This study used F4B ($\varepsilon_r$ = 2.65) with stable wave-absorbing properties as the dielectric substrate, and the thickness was h. Laying a metal patch on the lower layer of the resistive film structure can control the wave absorbing part of the resistive film structure to a certain extent to realize the stable and incomplete resonance wave absorption of the absorber.

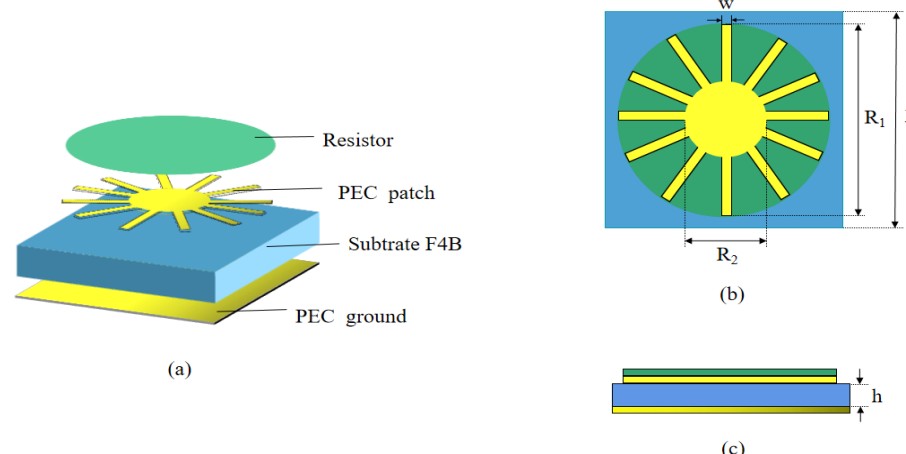

**Figure 2.** Resistive film periodic unit: (**a**) schematic structure; (**b**) top view; (**c**) side view (p = 5, w = 0.15, $R_1$ = 2.3, $R_2$ = 0.4, h = 0.5, unit: mm).

The simulation results of the resistive film absorbing structure (resistance value of 40 ohms) are shown in Figure 3. It can be seen from Figure 3a that the structure has good angular stability. The total reflection coefficients ($\Gamma_s$ is the product of the reflection coefficients of each face of the TCR) at 15 GHz and 35 GHz are −3.63 dB, −8.53 dB, comparable to the target's RCS reduction. As shown in Figure 3b, the wave absorption of the intermediate structure of the resistive film can be reduced to a certain extent so that the absorber can only achieve a certain proportion of wave absorption in a wide frequency band.

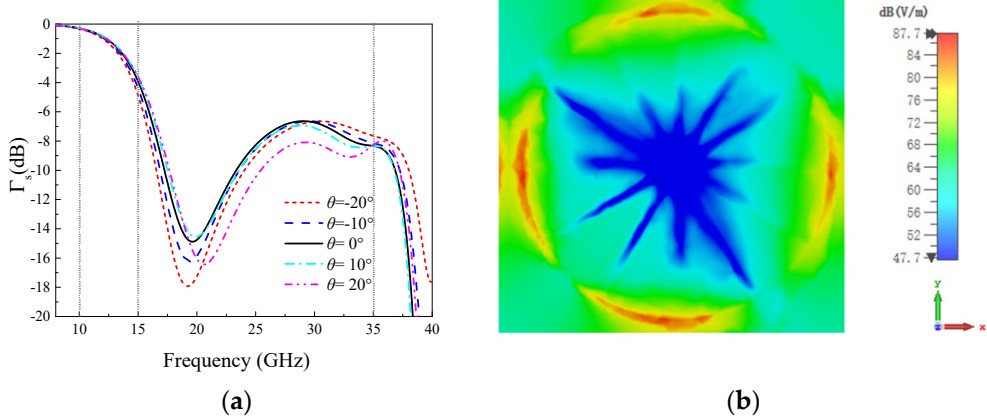

**Figure 3.** The resistive film periodic element: (**a**) total reflection coefficient; (**b**) current distribution at 35 GHz.

The TCR model of the loaded resistive film absorber is shown in Figure 4a. The edge length of TCR is a = 200 mm, and the absorber elements are evenly arranged on the three surfaces. Due to the thin absorber, the dielectric plate can be perfectly attached to the TCR without affecting the structural properties of the TCR itself.

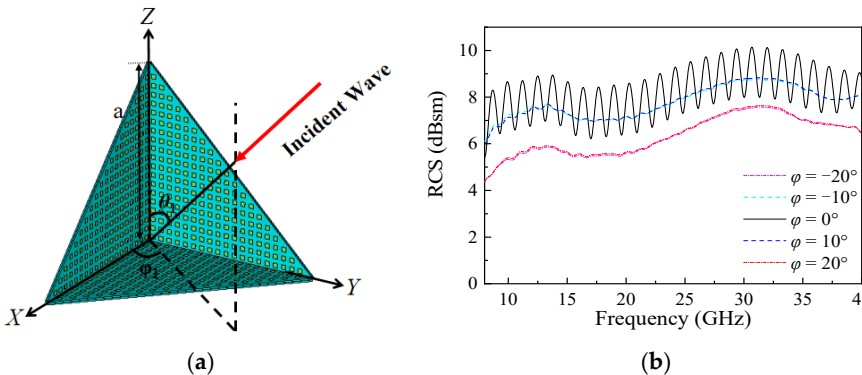

**Figure 4.** (**a**) Loading TCR of metasurface based on resistive film technology, (**b**) simulation results of the reflector at various incident angles of φ.

### 3.2. The Simulation Results of Metamaterial Surface Angle Reflector Based on Resistive Film Technology Are Loaded

The schematic diagram of the triangular TCR loaded on the metasurface is shown in Figure 4a (side length a = 200 mm), where the elevation angle and the azimuth angle are φ. When electromagnetic waves are "normally incident" to the TCR, $\theta_0 = 54.7356°$, $\varphi_0 = 45°$, and if the incident angle is shifted, the pitch angle and azimuth angle of the shifted normal incident are denoted as θ and φ ($\theta = \theta_1 - \theta_0$, $\varphi = \varphi_1 - \varphi_0$), respectively. The unit is a fully symmetrical structure; therefore, the effect of the change in φ on the RCS is only reflected in the change in the effective reflection area. Therefore, as shown in Figure 4b, the backward RCS waveform of the TCR remains unchanged, regardless of the change in φ. The simulations of the backward RCS values of the TCR loaded with the absorber based on the technology of the resistive film under different incident angles are shown in Figure 5. It can be seen from Figure 5 that no matter how the incident angle changes, the TCR can still maintain the backward RCS values at the center frequencies of the X, K, and Ka bands approximately equal.

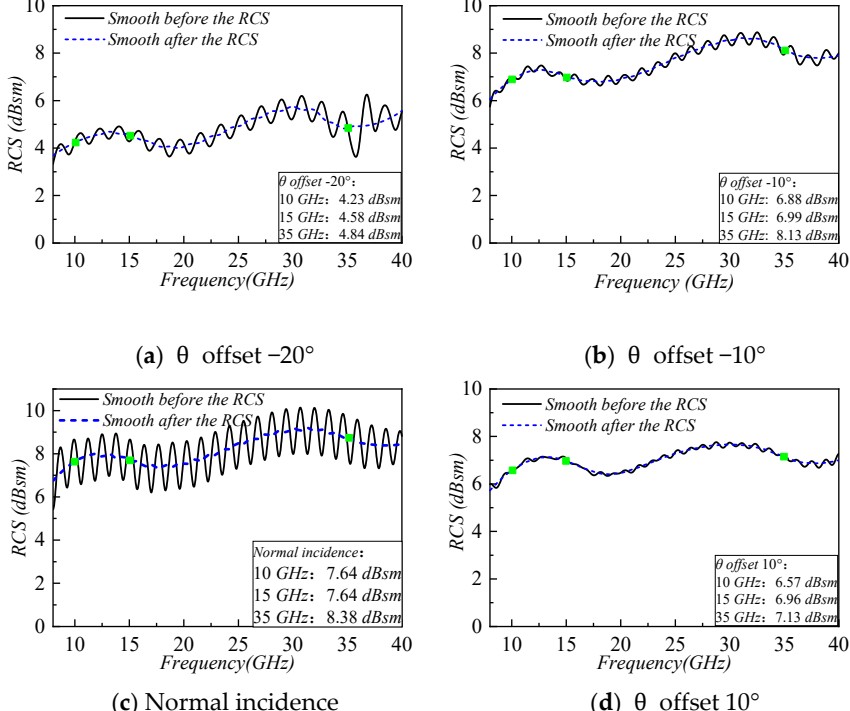

**Figure 5.** *Cont.*

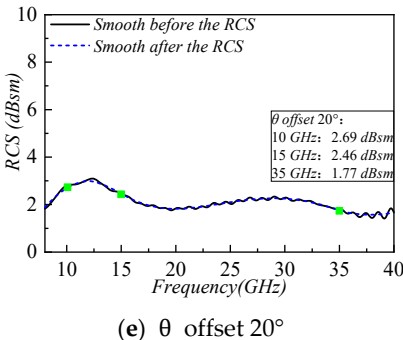

(**e**) θ　offset 20°

**Figure 5.** Simulation results of the loading TCR (side length a = 200) of the metasurface based on resistive film technology at various incident angles.

According to the simulation results in Figure 5 and Table 1, it can be seen that the TCR loaded with the absorber based on the resistive film structure has a good RCS reduction effect in the entire operating frequency band of 8–40 GHz under the condition of a large angle of ±20° oblique incidence. It can be realized that within the range of 1 dB fluctuation error, the RCS of X, K, and Ka bands center frequencies are roughly equal, and it has a good camouflage function compared with the ordinary TCR. The model of the machining test is shown in Figure 6. It can be seen from Figures 5 and 7 that the RCS image trend at normal incidence is the same as the test results and the simulation results. In the test results, the RCS result of 10 GHz is 1.39 dBsm less than the simulation result, the RCS result of 15 GHz is 0.07 dBsm more than the simulation result, and the RCS result of 35 GHz is 1.23 dBsm more than the simulation result. The test results are consistent with the simulation results when the resistance of the resistive film unit is 80 ohms; thus, there may be an inevitable error due to problems of processing accuracy of the resistive film. In general, it can be seen from the test results of normal incidence that the design is reasonable and feasible. Loading an absorber with features such as wide bandwidth and angular stability onto metal TCR can realize the precise control of TCR backward RCS in broadband multi-band and complete electromagnetic camouflage tasks in a wide-angle incident range.

**Table 1.** Net reduction values for different angles of inclination.

| The Angle of Inclination (θ) | Net Reduction of 15 GHz (dB) | Net Reduction of 35 GHz (dB) |
| --- | --- | --- |
| −20° | 0.35 | 0.61 |
| −10° | 0.11 | 1.25 |
| 0° | 0 | 0.74 |
| 10° | 0.39 | 0.56 |
| 20° | −0.23 | −0.92 |

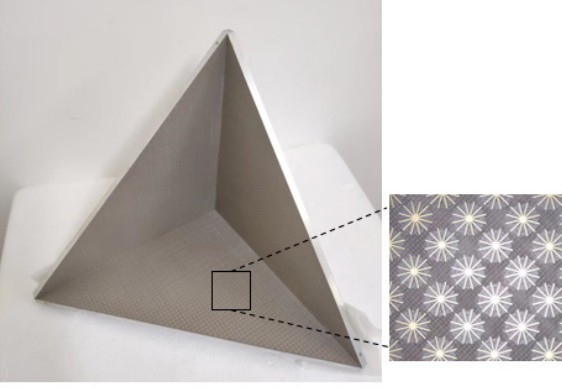

**Figure 6.** Machining TCR models.

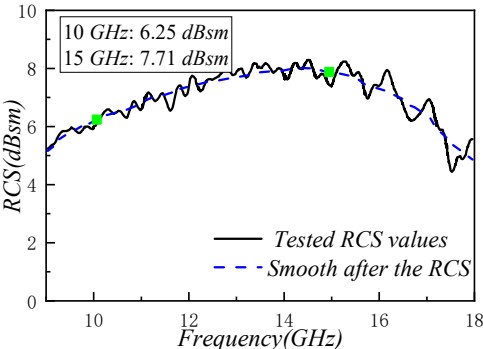 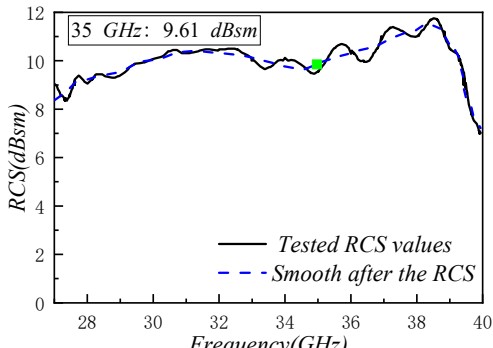

**Figure 7.** Test results of the normal incidence of TCR (side length a = 200) loaded on a metasurface based on resistive film technology.

## 4. Conclusions

This study aimed to overcome the control difficulties in cross-band corner reflectors. By studying the regulation mechanism under multiple scattering, a reflective absorbing metasurface based on resistive film technology was designed. The metasurface was loaded into the TCR, and the precise and flexible control of the RCS scattering properties of the strong backscattering target of the TCR was realized. Through experiments, the test results are in good agreement with the simulation results. This method solves the frequency instability of metasurfaces in the case of multiple reflections and provides a technical solution for the subsequent promotion of multiple reflection regulation of complex targets.

**Author Contributions:** Conceptualization, H.Y. (Hongcheng Yi); methodology, S.Z.; investigation, T.S. and C.X.; writing—original draft preparation, T.S.; writing—review and editing, F.Y. and H.Y. (Hua Yan); visualization, T.S. All authors have read and agreed to the published version of the manuscript.

**Funding:** This research was funded by the National Natural Science Foundation of China (62101515).

**Conflicts of Interest:** The authors declare no conflict of interest.

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
