# Peer review of "Ultrawideband Precision RCS Regulation for Trihedral Corner Reflectors by Loading Resistive Film Absorbers"

_electronics, doi:10.3390/electronics11223696_

Round 1
Reviewer 1 Report
Dear authors,
considering the topic interesting and well presented. I have been able to ascertain the knowledge of the subject and I think it is a good job.
The presentation of the results and also the conclusions should be improved. The bibliography must be implemented.
My studies on standard and large deployable satellite antennas could (perhaps) be an example of applications. Therefore you could cite the following publications:
- Cammarata, A., Sinatra, R., Rigano, A., Lombardo, M., & Maddio, P. D. (2020). Design of a large deployable reflector opening system. Machines, 8 (1), 7.
- Maddio, P.D., Meschini, A., Sinatra, R., Cammarata, A. An optimized form-finding method of an asymmetric large deployable reflector. Engineering Structuresthis link is disabled, 2019, 181, pp. 27–34.
- Maddio, P.D., Salvini, P., Sinatra, R., Cammarata, A. Optimization of the efficiency of large deployable reflectors by measuring the error around the feed. Acta Astronauticathis link is disabled, 2022, 199, pp. 206–223.
Thank you.
Regards
Author Response
Dear Editors and Reviewers:
Thank you for your letter and for the reviewers’ comments concerning our manuscript entitled “Ultrawideband Precision RCS Regulation for Trihedral Corner Reflector by Loading Resistive Film Absorbers” (ID: electronics-1895459). Those comments are all valuable and very helpful for revising and improving our paper, as well as the important guiding significance to our researches.
First, before we address the reviewers’ comments, we'd like to emphasize the contributions and the novelty of the method, which addresses the control difficulties in cross-frequency band and multi-polarization angle reflectors, and can realize the accurate and flexible regulation of the RCS scattering characteristics of strong backscattering targets such as TCR, so as to solve the problem of frequency instability of superstructure surface under multiple reflections. We are confident this is a useful contribution to filling the gap in the field of TCR scattering characteristic regulation using superconferential surfaces, and promoting the development of TCR in electromagnetic camouflage applications. Omitted comments will be fixed in revision if accepted.
We thank the reviewers for acknowledging the strong performance of this work and the quality of the presentation. We believe the method proposed is very effective and powerful. We agree with the reviewers that the details of the submitted manuscript should be improved and we have studied comments carefully and have made corrections which we hope meet with approval. Revised portions are marked in red in the paper. The main corrections in the paper can be found in the annex.
Sincerely,
Author

Reviewer 2 Report
This paper proposes an interesting structure for RCS regulation. However, the main goal of the discussed application is the RCS reduction and there is no convincing evidence (citation) that RCS regulation is, also, required. Moreover, the language manipulation in the paper is poor; thus, some points are not easily comprehensible. Some additional comments:
1) The final expression in (1) is a well-known relation that can be omitted. However, it is very strange that the limit of R to infinity in the expression 4*pi*R^2 is a finite value. Please explain this.
2) The reference to Figure 5 in line 88 is probably not correct.
3) In section 2, some simulation results are presented but there is not any information regarding the geometric setup (only the edge length is mentioned, while the equivalent setup is shown 3 pages later) and the numerical setup (number of mesh-cells, excitation type, etc.)
4) In 92-93, it is mentioned that "the net reduction of TCR backward RCS at 15 GHz and 35 GHz is 2.93 and 9.99". The reduction compared to what?
5) How is the expression in line 105 extracted?
6) The first paragraph in section 3.A does not make any sense. Please rephrase it.
7) The captions of some figures are on different pages (Fig. 2 and Fig. 5).
8) The authors refer to the polarization but it seems that only the angle of incidence is discussed. The polarization angle should be analyzed, too.
9) The authors claim that "the backward RCS waveform of the TCR remains unchanged regardless of the change in φ". However, in Fig. 4b there are significant changes. Please explain this.
10) The results in Figure 5, namely the RCS values at different frequencies, should be summarized in a compact form of a table.
Round 2
Reviewer 2 Report
The authors conducted several changes, and the paper seems to be improved.
1) However, there is not any mention concerning convincing evidence (citation) that RCS regulation is, also, required.
2) Moreover, the expression (1) still tends to infinity for R->infinity. Is this relation correct?
3) Regarding the change of phi, the pattern is, indeed, almost the same, but phi=10 to phi=20 has approximately a 2dB difference (30%). This is not a negligible value. Furthermore, concerning the fluctuation, an explanation for this behavior should be included.
4) The question for polarization does not focus on the possible polarization change, but in terms of polarization direction. Consequently, the authors must define what polarization direction and type is investigated. Moreover, the examination must extend to two orthogonal polarizations to be scientifically sound.
Round 3
Reviewer 2 Report
The question regarding the usefulness of RCS regulation is still not discussed appropriately.
Moreover, the authors must mention explicitly in the abstract and the introduction that the purpose of the work is the regulation of the RCS at a wide frequency band, not for different incidence angles (since the authors confirm that the 30% difference between different angles is rational).
An additional comment is raised in Table 1. Here, it is mentioned a net reduction for different angles. The reduction compared to what?
Finally, a comparison of all the aspects (RCS regulation and reduction) should be included with a resistive layer film-coated TCR (the entire surface). Indeed, this is a much simpler implementation, compared to the metamaterial surface, and can possibly provide better results than a purely metallic surface.
